# Current Views on the Interplay between Tyrosine Kinases and Phosphatases in Chronic Myeloid Leukemia

**DOI:** 10.3390/cancers13102311

**Published:** 2021-05-12

**Authors:** Christian Boni, Claudio Sorio

**Affiliations:** Department of Medicine, General Pathology Division, University of Verona, 37134 Verona, Italy; christian.boni@univr.it

**Keywords:** phosphatases, kinases, CML, leukemia, stem cell

## Abstract

**Simple Summary:**

The chromosomal alteration t(9;22) generating the BCR-ABL1 fusion protein represents the principal feature that distinguishes some types of leukemia. An increasing number of articles have focused the attention on the relevance of protein phosphatases and their potential role in the control of BCR-ABL1-dependent or -independent signaling in different areas related to the biology of chronic myeloid leukemia. Herein, we discuss how tyrosine and serine/threonine protein phosphatases may interact with protein kinases, in order to regulate proliferative signal cascades, quiescence and self-renewals on leukemic stem cells, and drug-resistance, indicating how BCR-ABL1 can (directly or indirectly) affect these critical cells behaviors. We provide an updated review of the literature on the function of protein phosphatases and their regulation mechanism in chronic myeloid leukemia.

**Abstract:**

Chronic myeloid leukemia (CML) is a myeloproliferative disorder characterized by BCR-ABL1 oncogene expression. This dysregulated protein-tyrosine kinase (PTK) is known as the principal driver of the disease and is targeted by tyrosine kinase inhibitors (TKIs). Extensive documentation has elucidated how the transformation of malignant cells is characterized by multiple genetic/epigenetic changes leading to the loss of tumor-suppressor genes function or proto-oncogenes expression. The impairment of adequate levels of substrates phosphorylation, thus affecting the balance PTKs and protein phosphatases (PPs), represents a well-established cellular mechanism to escape from self-limiting signals. In this review, we focus our attention on the characterization of and interactions between PTKs and PPs, emphasizing their biological roles in disease expansion, the regulation of LSCs and TKI resistance. We decided to separate those PPs that have been validated in primary cell models or leukemia mouse models from those whose studies have been performed only in cell lines (and, thus, require validation), as there may be differences in the manner that the associated pathways are modified under these two conditions. This review summarizes the roles of diverse PPs, with hope that better knowledge of the interplay among phosphatases and kinases will eventually result in a better understanding of this disease and contribute to its eradication.

## 1. Introduction

Protein phosphorylation is a fundamental mechanism for many intracellular processes underlying cell life [1]. This reversible mechanism, which is triggered by intra- and extra-cellular signals, regulates metabolism, transcription, proliferation, differentiation, cell movements, and apoptosis in countless cellular functions. Protein kinases form an enzyme family that catalyzes the transfer of the gamma-phosphate of adenosine triphosphate (ATP) to specific hydroxyl amino acids in protein substrates, in particular on serine (S), threonine (T), or tyrosine (Y) residues [2]. Phospho-group integration cause structural modifications and functional effects which are instrumental in the transmission of appropriate signals. In mammalian cells, protein kinases take up 2% of the genome and are one of the largest families of genes, coding for over 500 proteins [3]. Human protein tyrosine kinases (PTKs) can be categorized into about 60 trans-membrane receptors (PTKRs) and 30 non-receptor classes (PTKNRs) [4]. PTKRs may be activated by specific molecular cues, upon binding to their extracellular ligand-binding domains. These events usually induce intracellular structural reorganization, increasing affinity for substrates, and specific sub-cellular localization, to allow for the initiation of intracellular signals [5]. In contrast, PTKNRs lack transmembrane domains, localize in the cytoplasm, and are characterized by different domains, including kinase domain (KD), Src homology domains (SH2 and SH3), or other ligand-binding domains such as phosphotyrosine-binding (PTB) domains, involving tasks such as signals propagation and amplification [6,7]. PTKs have been predominantly implicated in cancer development, although these targets have also been considered in the treatment of illnesses such as hypertension, Parkinson’s disease, and autoimmune diseases [8]. Aberrant PTK activation in cancers is currently known to be mediated by four principal molecular mechanisms: Autocrine stimulation, chromosomal translocations, overexpression of PTKRs, or gain of-function mutations [9]. A commonly reported example is the juxtaposition of *BCR* on the *ABL1* gene, causing the transduction of BCR-ABL1 fusion protein, a very well-studied state of kinase-alteration. The gene region involved in the translocation may generate different isoforms of the BCR-ABL1 protein, which are often associated with a specific leukemic phenotype. Unlike the Src-family kinases, where two specific phosphorylation sites (corresponding to Y416 and Y527 of the archetype SRC) control the enzyme activity with contrasting effects, the ABL family does not contain an inhibitory site but, instead, self-inhibited states are regulated by amino-terminal myristoylation of the SH2 and SH3 domains, both contributing to an inactive kinase domain conformation [10]. Moreover, ABL kinase activity depends on the tyrosine residues located on its structure, which are substrates for phosphorylation events that stabilize the open-active conformation [11]. In the fused conformation, BCR coiled-coil domains promote intermolecular cross-phosphorylation on ABL kinase, enhancing its kinase activity and promoting a constitutive activated tyrosine kinase [12]. (Figure 1 [13,14]).

Myeloproliferative neoplasms are clonal disorders propagated by transformed hematopoietic stem cells (HSCs). Chronic myeloid leukemia (CML) is one such disorders, which is characterized by a HSC mutation involving an exchange of genetic material between chromosomes 9 and 22, usually t(9;22)(q34;q11) traslocation [15]. This translocation generates a cancer-driving gene known as *BCR-ABL1*, coding for a dysregulated protein tyrosine kinase. Over the last 20 years, innovative cures for many common cancer types have been developed based on therapies targeting tyrosine and serine/threonine kinases. In CML, TKI-based therapy is considered the gold standard for treatment, having dramatically changed the clinical outcome since the first TKI—imatinib or Glivec^®^ (IM)—was approved for clinical use in 2001 [16]. Since then, numerous efforts have been put forward to counter various BCR-ABL1 mutations occurring in the active kinase site, with the development of second- and third-generation TKIs over the past decade [17]. Despite the success of these inhibitors (e.g., Dasatinib, nilotinib, and bosutinib), even in contrasting the “gatekeeper” T315I mutation (ponatinib), the uncomfortable truth from numerous clinical trials indicates that the success of TKIs in achieving clinically relevant endpoints is still not optimal [18,19,20], suggesting the need to tackle additional mechanisms to achieve what is becoming a foreseeable goal: the eradication of CML. Meanwhile, further strategies are needed to prolong the survival of leukemia patients who are resistant or efractory to current chemo- and TKI-based therapies.

Copious evidence has supported the role of kinase oncogenes in cancer development. Nevertheless, there has been a heightened interest in the pivotal role of specific protein phosphatases in hematological malignant transformation processes. The human PTPome contain 107 PTPs, which are grouped into four families based on their catalytic domain amino acid sequence. Unlike protein tyrosine phosphatases (PTPs), proteins with serine/threonine phosphatase activity are encoded by only 13 genes [21]. However while PTPs are monomers, the serine/threonine phosphatases (e.g., PP2A) are multimeric and include various regulatory sub-units, resulting in hundreds of different isoforms, which allows for the substrate specificity necessary to target thousands of phosphoproteins. The majority of oncogenes identified thus far encode protein kinases, the activity of which is required for cancer initiation and maintenance. Protein phosphatases (PPs) often counteract the action of protein kinases by removing phosphate moieties on target proteins [22,23]. Intuitively, solely considering the counterbalancing activity of protein kinases, it is possible to consider PPs to act as tumor suppressors; however, this concept represents an oversimplification. Deregulation of or changes in the expression/activities of phosphatases might tip the overall cellular homeostasis, establishing one of the mechanisms by which cells escape external and internal self-limiting signals, thus resulting in altered cellular processes [22]. Although much is known about the alterations in kinase function in CML disease, the roles of phosphatases in these same processes remain less defined and represent a matter of great interest, as the recovery of tumor- suppressor function represents a challenge in cancer treatment. Strengthening our knowledge on the biological aspects of the phosphatases and their regulation of oncogenic processes in hematological malignancy will allow us to obtain better results in our understanding of molecular networks. This review summarizes the current knowledge on the biological roles played by protein phosphatases in CML, leading the discussion on their interactions with protein kinase and the functional aspects that rules these PPs. 

For this review, we decided to separately discuss PPs for which data on CML primary cells or leukemia mouse models are available as a cellular model of disease, compared to cell lines that are known to develop cell culture-associated heterogeneity/alterations, which might not entirely replicate the molecular events present in primary cells or in vivo conditions. Moreover, we highlight their key role in the self-renewal and proliferation of CML cells, as well as in TKI resistance phenomena. We hope that these remarks can assist in developing a better understanding of critical BCR-ABL1 downstream pathways, which may represent attractive therapeutic targets to develop TKIs supporting therapies, in order to overcome the functional alteration of these phosphatases.

**Figure 1 cancers-13-02311-f001:**
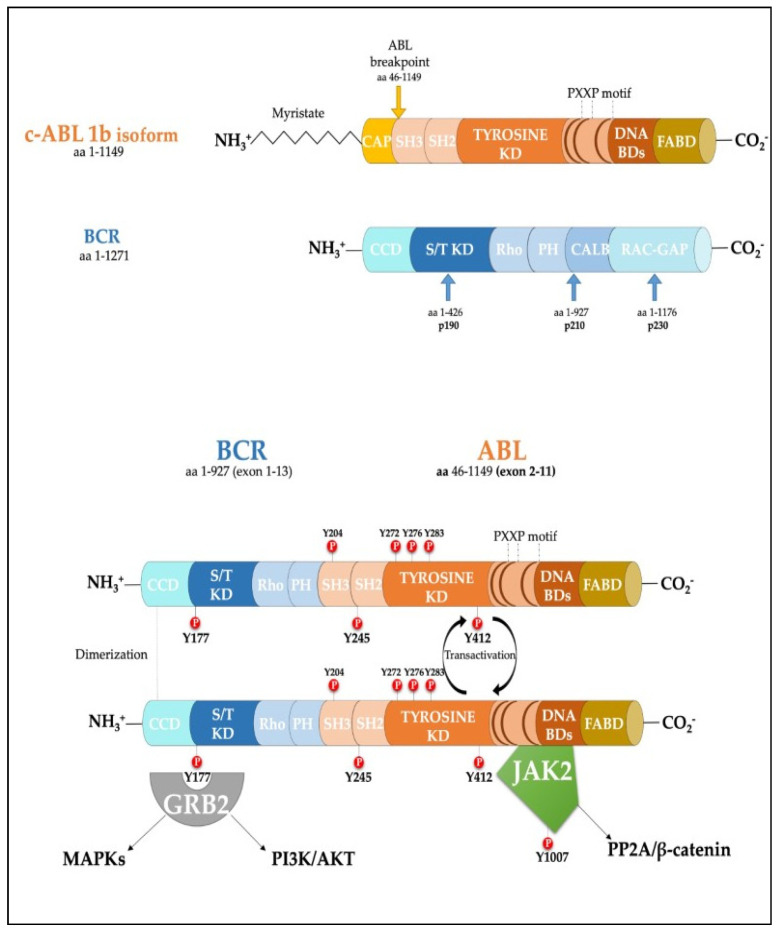
Scheme of the protein products derived from the BCR, ABL1b genes and the fusion product of the BCR-ABL1 oncogene. Major tyrosine phosphorylation sites are represented together with two well characterized interactors relevant for CML disease [13,14]. SH Src homology domain; PTB Phosphotyrosine-binding domains; KD Kinase Domain; PH Pleckstrin homology domain; BD Binding Domain; FABD f-actin binding domain; CCD Coiled-coil domain.

## 2. Role of Phosphatases in Chronic Myeloid Leukemia

### 2.1. Role of Phosphatases in the Regulation of Cell Proliferation

In the last three decades, the role of the BCR-ABL1 protein has been extensively analyzed and we have summarized the main pathways involved in Figure 2. One of the major signaling cascades that is altered involves the RAS mitogen-activated protein kinase family (MAPKs) [24], such as ERK1/2, MEK, JNK, and p38 families, which lead to insensitivity to growth factor stimuli and regulates the proliferative fate of cells [25,26]. In the active conformation, BCR-ABL1 possesses several tyrosine sites, such as Y245 in the SH2-kinase linker and Y412 in the activation loop, which are known to regulate function/activity (Figure 1). The human *UBASH3B* gene encodes for STS-1 phosphatase (suppressor of T-cell receptor signaling 1), the high expression of which has been correlated with the p190^BCR-ABL1^ form in acute lymphoblastic leukemia (ALL) patients’ samples [27], but is usually considered a BCR-ABL1 interactor [28,29]. In particular, other than being involved in signaling by other kinases, such as PDGFR, ZAP-70, and SYK, it can bind to the SH2–SH3 compartment of BCR-ABL1, producing strong dephosphorylation in the tyrosine sites of the ABL-portion, thus limiting its kinase activity. Remarkably, a CML-like disease mouse model with *Sts-1/Sts-2* double knockout exacerbated the classical CML parameters and reduced mice survival, providing evidence supporting the leukemogenic role of Sts-1 [30]. The SH2-domain containing protein growth factor receptor bound protein-2 (GRB2) recognizes another fundamental Y177 binding site on the BCR domain which synergistically promotes CML, supporting RAS activation through Son of Sevenless (SOS) a guanine nucleotide exchange factor protein, and the scaffold adapter GAB2 (GRB2-associated binding protein 2) [31]. BCR-ABL1^+^ cells harboring a Y177F or SH2-domain mutation on GRB2 have exhibited less leukemogenic transformation and reduced proliferation by MAPK pathway disruption [32,33]. A recent study has also suggested that Y177 BCR-ABL1 phosphorylation is also regulated by ERα36-expression, an alternatively spliced variant of estrogen receptor α66 (ERα66), which is abnormally localized in the cytoplasm and cell membrane of BCR-ABL1^+^ cells. Synthetic ERα36 inhibitors prevented GRB2 from binding to BCR-ABL1, generation reduction in the downstream RAS/MAPKs pathway and cell proliferation impairment [34]. The major SHP2-binding protein in hematopoietic cells, GAB2, is part of the GRB2/GAB2 complex recruited on phosphorylated Y177. GAB2 is essential for myeloid and lymphoid leukemogenesis induced by BCR-ABL1, as demonstrated by the failure to develop a CML-like disease in a mouse model transplanted with BCR-ABL1 in *GAB2^−/−^* marrow cells. Its tyrosine phosphorylation in BCR-ABL1^+^ cells generates a docking site for signal cascade proteins, such as PI3K sub-unit p85α and SHP2 [35]; moreover, it induces the signal events, together with RAS activation, that lead to an increase of ERK function [32]. Downstream proliferative effects of the MAPKs pathway include transcription factors (TFs) such as NF-κB, CREB, ETS-1, AP-1, and cMYC, leading to cell cycle progression (involving CDKs) and anti-apoptotic mechanisms (involving BCL-2) [36]. SHP2 (*PTPN11*) plays a critical role in cell development due to its ability to support RAS/MAPK signaling, in response to numerous growth factors [37,38]. In the hematopoietic compartment, it is considered an oncogene as point mutations in its N-terminal SH2 inhibitory domain trigger the development of leukemia in different lineages [39]. An early study performed in CML CD34^+^ cells reported that the cytokine-independent colony formation capacity was altered by *PTPN11*-knockdown [40]. In particular, both mRNA and phosphoprotein levels appear to be greater in myeloid leukemia cells, addressing its presence/activity with a more hyper-proliferative phenotype [41]. Further evidence has confirmed SHP2 is required to initiate and maintain BCR-ABL1-mediated transformation, as GAB2 mutation in SH2 domain cannot bind SHP2 and, as a result, reduce myeloid and lymphoid leukemic burden. In line with this observation, various studies have described p-ERK activity reduction in BCR-ABL1 expressing myeloid cells lacking SHP2 phosphatase, thus emphasizing its role as a positive regulator of the RAS/MEK/ERK1-2 pathway in BCR-ABL1 signaling [32,42,43,44]. Speculation on reduced cell viability, in this case may be traced back to RAS activation which occurs through SOS, but which also requires active SHP2 phosphatase. SHP2 can dephosphorylate and inactivate the p120 RAS-GAP protein (RAS GTPase-activating protein) by blocking its antagonistic action on RAS [38,45]. Although there are also implications for p-AKT-reduction, it seems to be mostly associated with GAB2 adapter protein impairment. On the other hand, Y177-GAB2 interaction is required to enforce another key player in leukemic transformation: Phosphoinositide-3 kinase (PI3K) [32]. The PI3K/AKT signal pathway plays a crucial role in a great variety of cancers, due to its strong relation with membrane tyrosine kinase receptors which are activated by multiple extracellular stimuli (cytokines, growth factors, etc.) [46]. Reportedly, the PI3K lipid kinase consists of p85 (regulatory) and p110 (catalytic) sub-units in a heterodimeric structure that is involved in inositol lipids (PtdIns(3,4)P2) phosphorylation, usually at the inner leaflet of the cytoplasmic membrane, initiating and controlling multiple cellular functions [47]. Specifically, PtdIns(3,4,5)P3 provides an anchor point for pleckstrin homology (PH) domain-containing proteins, such as activated protein kinase (AKT) and phosphoinositide-dependent protein kinase-1 (PDK-1). AKT is a serine/threonine kinase, belonging to the AGC family kinases, which is expressed in two isoforms in hematopoietic stem cells [48]. Activated p-AKT following by BCR-ABL1-activated PI3K, is able to inhibit the apoptotic process and support cell proliferation [49,50]. Skorski et al. reported the first direct data on PI3K (both sub-units) involvement in CML cell transformation [51], demonstrating how its downstream effector, AKT, was a critical growth regulatory switch in BCR-ABL1-expressing cells [52]. BCR-ABL1 can directly up-regulate AURK-A and AURK-B (serine/threonine kinases belonging to the Aurora family) through (at least in part) AKT, emphasizing a further pathway for cell, proliferation [53]. The PI3K pathway is constitutively activated in CML progenitor cells, as has been demonstrated by the elevated PtdIns(3,4,5)P3 levels found in CML progenitor cells, compared to their normal counterpart. Normal levels of PtdIns(3,4,5)P3 were restored, along with their capacity to respond near-normally to cytokine stimulation, by Imatinib treatment [54]. Many studies on this signaling pathway have been deepened in CML biology, thanks to the development of specific inhibitors to curb a wide range of effectors acting on this leukemia [55,56]. Notoriously, phosphatase and tensin homologue (*PTEN*) was the first tumor suppressor gene found to have a double specific phosphatase activity, acting on serine/threonine and tyrosine residues and the major antagonist of PI3K signaling, operating as a tumor suppressor in numerous solid neoplasms and leukemia [57,58]. By exploring the function of PTEN in BCR-ABL–expressing Ba/F3 cells, it has been revealed how BCR-ABL1 may induce PTEN phosphatase downregulation and a reduced p53 protein expression, regaining expression of both targets after TKI treatment [59,60]. An explanation was found in *Pten* gene promoter, where p53 can bind and promote its expression. Additionally, BCR-ABL1-expressing LSK cells sorted from CML mice, confirm a reduction of Pten mRNA, which also correlates with the downregulation of p53. Therefore, reduced disease progression related to PTEN expression could involve the reduction of CML cell proliferation through cell cycle arrest [59]. The nuclear-cytoplasmic shuttling enables PTEN to play its proper tumor suppressive function [61]. The mono-ubiquitinated PTEN form is predominantly nuclear, where it has been shown to play a tumor-suppressive role through a proliferation control mechanism. Herpesvirus ubiquitin-specific protease (HAUSP) is a critical modulator of protein ubiquitination, possibly regulating PTEN cytosolic partitioning [62]. In CML, BCR-ABL1 might phosphorylate HAUSP, triggering PTEN nuclear exclusion and causing proliferative advantages. Therefore, HAUSP together with nuclear PTEN depict a pivotal pathway in BCR-ABL1-induced proliferation [63]. In this way, Morotti et al. showed that PTEN-tail phosphorylation, mediated by BCR-ABL1-activated casein kinase II (CKII), caused a reduction of PTEN phosphatase activity, thus uncovering another inhibition mechanism present in CML [64]. Many different mechanisms might regulate PTEN expression in leukemic cells, including epigenetic shutdown, genomic loss, transcriptional repression, post-transcriptional regulation by lncRNA or microRNAs, etc. Polycomb repressive complex 2 (PRC2) contains two methyltransferase sub-units called enhancers of zeste 2 polycomb repressive complex subunit 1 and 2 (EZH1 and EZH2) that can modulate chromatin conformation by adding methyl groups on histone 3 (H3K27me2/3) [65]. EZH2 expression has been found to be upregulated in all three-phases of the disease in CML LSCs and, in particular, compared to other PRC2 components, it appears to be BCR-ABL1-kinase activity-dependent; meanwhile, EZH1 did not follow this trend, but was found to be downregulated [66,67]. Proliferation, self-renewal and viability of CML cells were drastically impeded by pharmacological interference with EZH2 activity (GSK126 inhibitor), or by reducing its expression through shRNA or in *Ezh2^−/−^* CML mice, resulting in increased survival in retroviral *BCR-ABL1*-transduced mouse models [66,68]. In addition, *EZH2* knockdown decreased H3K27me3 levels on *PTEN* gene promoter and resulted in a significant increase in PTEN mRNA and protein expression in three human Ph^+^ cell lines, as well as in LSK (Lin^−^/Sca-1^+^/c-Kit^+^) cells from CML mice. Zhou et al. demonstrated the EZH2 inhibition-mediated beneficial effects on leukemia cells and prolonged survival of CML mice were compromised by the concurrent transduction of shRNA targeting *PTEN* [68]. Modification of PTEN expression might occur through maternally expressed gene 3 (MEG3), a long non-coding RNA (lncRNAs) associated with many cancers which has already been shown to regulate IM resistance in CML [69]. Lower MEG3 levels have been found in advanced stages of leukemia while, to the contrary, miR-21 was found to be over-expressed which was able to interact with MEG3 by decreasing its expression. The data indicated that MEG3 binding could modify the expression of MDM2 and EZH2 mRNA levels, producing DNA (cytosine-5)-methyltransferase 1 (DNMT1) protein upregulation and PTEN protein downregulation.

Therefore, a reduced expression of miR-21 blocked the proliferation and promoted apoptosis of CML cells [70]. The upregulation of miR-188-5p represents another post-transcriptional regulatory mechanism for PTEN expression exhibited by CML cells. Indeed, miR-188-5p may directly target PTEN 3′-UTR, thus repressing it. Zi-Yuan Nie et al. demonstrated how the flavonoid Morin, isolated from *Moraceae*, might inhibit proliferation and induce apoptosis by repressing miR-188-5p expression, leading to PTEN/AKT pathway inhibition in CML cells both in the K562 cell line and mouse xenograft models [49]. Yin et al. obtained relevant data on other epigenetic modifications produced by the RBP2 protein. In particular, the BCR-ABL1/RBP2/PTEN pathway represents a feedback loop which is thought to increase proliferation, leading to blast phase (BP) transition. BCR-ABL1 activity directly inhibits directly RBP2 protein expression, resulting in the inhibition of PTEN transcription. The PTP domain of PTEN shares similar features with PTP1B phosphatase, presenting the same ability to bind and limit BCR-ABL1 phosphorylation [71] (see Figure 2).

### 2.2. Role of Phosphatases in Regulation of Quiescence and Self-Renewal in Progenitor and Leukemia Stem Cells (LSC)

Despite having been proposed decades ago, LSC theory is still considered a robust framework for our understanding of leukemia. This theory has informed the hierarchical organization of leukemic (sub)clones: more mature cells enter an apoptotic state after a number of cell divisions, while missing cells are replenished by LSCs featuring unlimited self-renewing capability, thus ensuring leukemia-propagating capacity [72,73,74]. Given these premises, it is obvious that LSCs are considered the major target of any therapy aimed at leukemia eradication, and countless studies have aimed to understand the life cycle of this tiny population, which leads to insensitivity to TKI therapies and disease relapse [75]. Despite this, the exact nature of CML LSCs are still elusive. CML initiation and propagation have been associated to BCR-ABL1-dependent or -independent pro-survival and self-renewal misshapen pathways. Evidence from the literature has indicated that BCR-ABL1 in the hemangioblast may contribute to both malignant hematopoiesis and endotheliopoiesis [76]. In this regard, better characterization of the quiescent cell populations that define this disease is highly desirable. The terms leukemia stem cells (LSCs) and leukemia initiating cells (LICs) have been widely used to define the cell populations that need to be targeted to achieve a definitive cure for any form of leukemia, including CML [77]. While there is agreement on the surface phenotype defining CML LSCs (CD34^+^/CD38^−^/CD90^+^/CD26^+^), which might also include CD25^+^, IL-RAP1^+^ additional markers such as CD9, CD33, CD123 and CD56 were recently proposed [78,79,80]. At least some LSCs may also reside in CD34^+^/CD38^+^ subsets or, even, in CD34^−^ sub-fractions [81,82]. Different populations with LICs capability have been reported to be responsible for the relapse of the disease due to the acquisition of further cellular abnormalities induced by the selection pressure associated to the treatment with TKIs (e.g., the T315I mutation on the ABL domain) [83]. Both undifferentiated and more differentiated CML progenitor cells have been related to numerous signaling molecules including several phosphatases. An example has been reported regarding the function of Pten in mouse bone marrow LSKs (Lin^−^/c-Kit^+^/Sca1^+^). Specifically, these cells transduced with Bcr-Abl1, in combination (or not) with Pten, were transplanted into recipient mice. Flow cytometry analysis after 14 days indicated that the LSKs Bcr-Abl1/Pten-transduced cell number was significantly lower, compared to those transduced only with Bcr-Abl1, indicating a controlling role of Pten in mouse LSKs function [59]. Since the canonical WNT signaling pathway is one of the pathways involved in cell-differentiation, multiple effectors tightly regulate it. The altered balance between PPs and PTKs represents an essential step in β-catenin signaling during CML progression [84]. During blast-stage, CML-increased BCR-ABL1 expression leads to high levels of nuclear and cytoplasmic β-catenin [85,86]. These high levels are due to its stabilization, due to its changed degree of phosphorylation, representing an indicator of CML progression. One of the BCR-ABL1-independent checkpoints acting on β-catenin functions occurs through the interactions of JAK2 and PP2A, as well as other direct interactions take that part in regulation, such as AURK-A [87]. JAKs family members are key pathogenetic players in the induction [88] and evolution [89] of CML, representing critical therapeutic targets in CML LSCs [90]. JAK2 represent a crucial PTK, as it physically binds the C-terminus of BCR-ABL1, whose SH2 domain allows JAK2 tyrosine phosphorylation and activation [91]. Likewise, JAK2 directly phosphorylates the pivotal Y177 residue on BCR-ABL1, modulating several important pathways such as RAS/MAPK and PI3K [92]. Another player of relevance for transformation is PP2A, whose genetic alteration or loss of function define common adverse events shared by several human cancers and leukemia. This serine-threonine phosphatase regulates cell proliferation, survival, and differentiation, serving as a critical regulator of WNT signaling by directly modifying stability of β-catenin or acting on GSK3-β (glycogen synthase kinase 3-β) and AKT, which are upstream controllers of β-catenin [93,94,95]. Originally Neviani et al. described a BCR-ABL1-mediated upregulation of SET (I2PP2A, inhibitor 2 of PP2A) through hnRNP-A1 induction, occurring simultaneously with PP2A subunit phosphorylation on Y307. These events are enforced by BCR-ABL1 itself, or by other kinases (e.g., JAK2); all of these mechanisms are underpinned by PP2A activity suppression in CP and BP-CML primary progenitor cells [96]. Moreover, PP2A inhibition increases BCR-ABL1 phosphorylation through the downregulation of BCR-ABL1-associated SHP1 phosphatase, which protects the fusion protein itself from proteasome degradation [96]. Concurrently, JAK2 acts downstream of BCR-ABL1, by upregulating SET expression, which activates the LYN kinase through the SET-PP2A-SHP1 pathway [97]. In line with these results, SET-siRNA treatments, JAK2 inhibition, and PP2A reactivation with specific PADs (PP2A-activating drugs) such as FTY-720 or Forskolin support the critical role of PP2A reactivation in CML leukemogenesis [97,98]. Along the PP2A pathway, the cancerous inhibitor of protein phosphatase 2A (CIP2A) protein has been found to be a relevant player in CML BP progression, as has previously been demonstrated in other malignancies. Specifically, CIP2A may promote oncogenesis in CML, displacing the balance of pro-/anti-apoptotic proteins towards an apoptosis inhibition profile [99,100]. Recently further developments involving the role of PP2A in both LSC and Ph^+^ committed progenitor cells have been derived from the description of the complex regulatory activity carried out by *MIR300*, in a differential target-choice and dose-dependent manner (Figure 3). The *hsa-miR-300* (*MIR300*) is an intergenic miRNA belonging to *DLK1-DIO3* a genomic region located on human chromosome 14 (14q32) [101]. This region expresses maternal or paternal genes, depending on the allelic silencing regulated by methylation processes based on parental imprinting. *DLK1-DIO3* contains the largest miRNA cluster in the human genome with 54 miRNAs [102]. One of these is *MIR300*, produced not only by CML cells, but also by BM-derived primary MSCs, conferring a regulatory potential at the marrow niche. Effective levels of *MIR300* expression have been detected to be maintained in LSCs (CD34^+^/CD38^−^), affecting *CCND2* and *CDK6*, as predicted, and validated target genes, thus inducing their downregulation, consequently causing cell cycle arrest and promoting quiescence. Meanwhile, higher *MIR300* levels than those needed to block *CCND2* and *CDK6* are able to downregulate the expression of other effectors, including *SET*, *JAK2,* and *c-MYC,* resulting in PP2A-reactivation and inducing apoptosis. This does not occur in CP and BP CML progenitor cells, where BCR-ABL1 further inhibits *MIR300* expression levels, protecting them from apoptosis and cell cycle arrest [103]. Other PTPs play controlling roles in β-catenin protein functions. The *PTPN13* gene known as Fas-associated phosphatase-1 (FAP-1) acts as a repressor of Fas-induced apoptosis [104,105]. Interferon consensus sequence-binding protein (ICSBP) downregulates *PTPN13* through its *cis-element* promoter region, which is expressed in B-lymphoid and myeloid cells; however, it has been implicated in differentiation only in the latter, also acting a regulator of cell proliferation [106]. In particular, low levels of ICSBP have been found in primary CML BM cells [105,107]. FAP-1 interacts with the PDZ domain of the FAS protein, in order to mediate apoptosis [104]. Similarly, this phosphatase binds with the C-terminal of adenomatous polyposis coli (APC), a member of multiprotein complex coupled with AXIN and GSK3-β, which regulates β-catenin ubiquitination [108]. Strikingly, retroviral Bcr-Abl1 expression in murine myeloid progenitor cells mediated Icsbp downregulation, which lead to increased levels of Fap-1 binding to Apc, thus causing tyrosine-dephosphorylation and decreasing Gsk-3β activity. The inhibition of Gsk-3β reduces serine/threonine phosphorylation of β-catenin, facilitating its ubiquitination and consequent proteasome degradation [109]. These Bcr-abl1-related effects have been overthrown by SLV peptide treatment blocking the Fap-1/Apc interaction [109,110]. β-catenin has recently been found to be the target of another phosphatase, PTP receptor type γ (*PTPRG*), which was originally identified as a candidate tumor suppressor gene in renal cell and lung carcinoma, localized at 3p21-p14, a genomic region which is commonly deleted in these neoplasms [111]. PTPRG has been found to be expressed in myeloid cells including CD34^+^ precursors [112], where it plays an appreciable role either in hematopoietic differentiation or in the regulation of JAK2 within integrin-mediated monocyte adhesion [113,114,115]. Della Peruta et al. described the interaction and consequent downregulation of BCR-ABL1 kinase activity, thus first establishing its tumor suppressive role in CML. PTPRG expression (both at mRNA and protein levels) appears to be inversely correlated with the proliferative and clonogenic capacities of CML cells, and represents a disease-associated biomarker [116,117]. Recently, Tomasello et al. have proposed a mutual interplay between PTPRG and BCR-ABL1, involving β-catenin upregulation. This regulative mechanism requires the ability of PTPRG to dephosphorylate BCR-ABL1 on its SH2–kinase-domain linker, thus damping the full activation of the kinase. This limitation of BCR-ABL1 prevents complete β-catenin phosphorylation, with consequent destabilization and proteasome degradation. In addition, PTPRG seems to directly dephosphorylate β-catenin, providing further details on its tumor-suppressive effects in CML [118]. Methylation play an extremely important role in the regulation of PTPRG expression in CML—a feature shared with other malignancies [119]. In CML, promoter and intron-1 sequences of the *PTPRG* gene are consistently hyper-methylated by DNMT1/DNMT3-b upregulation, due to increased β-catenin stability [118,120]. Many studies mentioned above have underlined how multiple processes involving PPs converge to stabilize and activate the WNT/β-catenin cascade becoming part of the CML BP progression and pathogenesis.

### 2.3. Role of Protein Phosphatases in the Modulation of TKIs Response and Resistance Mechanisms

The clinical application of tyrosine kinase inhibitors (TKIs) radically changed the disease prognosis of CML, even if TKI resistance may suggest BCR-ABL1-independent signals sustaining a cellular transformed-clone which is able to drive disease recurrence [121]. Such driving mechanisms include BCR-ABL1 kinase domain mutation [122], increased reactive oxygen species (ROS) inducing DNA alterations [123], influx/efflux pumps abnormalities, etc. [124,125]. In this chapter, we focus our attention on several studies addressing the expression modifications or functions of PPs occurring in CML cells which are capable of promoting TKI insensitivity (see Figure 4). There has been mounting evidence that kinases and phosphatases participate in the drug-resistance processes in CML, assisted by micro-environmental factors or by MSCs-induced TKI protection [126,127]. For instance, JAKs signaling molecules regulate CML LSC, involving their niche protection from TKIs [128]. The bone morphogenetic proteins (BMPs) are critical players in the biology of stem cells (SCs), including HSC and cord blood progenitor cells. Notably, BMP4—together with a non-canonical JAK2/ signal transducer and activator of transcription 3 (STAT3) pathway—seems to directly regulate CML LSC quiescence. The BMP4-producing MSCs are closely correlated with a portion of LSCs (adherent), favoring the cell cycle exit of the latter through the BMP4 binding to the receptor serine/threonine kinase BMPR1B. The dormant state of such LSCs is also enhanced by TKI-treatment, resulting in drug resistance. Reportedly, a new therapeutic strategy using a mix of inhibitors (e.g., AG490, a JAK2 inhibitor, and E620, BMPR1B inhibitor) has been proposed, which the goal of reversing this dormant cell-state and promoting proliferation and TKI responsiveness [129]. Similarly, an unclear mechanism, involving the unusual secretion of IL-7, mediated by MSCs, caused IM and Nilotinib resistance through the JAK1/STAT5 pathway. Indeed, cytokine limitation or JAK pathway inhibition may assist TKIs to overcome such niche protection [130]. A recent study by Mancini et al. has reported the hyper-activation of the Aurora kinase A, which is an important target for both cell proliferation and quiescence [53]. This serine-threonine kinase is associated with the PLK1-FOXM1 axis supporting the leukemic progression and enabling cell death evasion in IM-resistant BCR-ABL1^+^ cells [87], but not in ponatinib-resistant cells [131]. This underlines how the TKI-resistance processes are highly variable in BCR-ABL1^+^ cells. As described for kinases, the modulation of also PPs affects the TKI-response, through external growth factor stimuli. Particularly, an analysis of gene-expression profiles was performed in different BCR-ABL1-expressing cell lines and in primary CML bone marrow cells after treatment with a combination with both growth factors and TKI. Comparison of the data sets revealed the modulation of two specific genes, c-*FOS* (belonging to the AP1 family) and dual-specificity phosphatase-1 (*DUSP1*). Notably, DUSP1 is a nuclear PP that act both on serine/threonine and tyrosine residues involved in MAPK signaling, specifically by interacting and inactivating ERK, JNK, and p38 MAPKs [132]. Specific deletion of *c-Fos* and *Dusp1* in an IM-treated CML mouse model induced substantial elimination of all leukemia cells without interfering with normal hematopoiesis. Furthermore, the pharmacological inhibition of two target proteins, together with BCR-ABL1, using IM + difluorinated curcumin (DFC) + (*E*)-2-benzylidene-3-(cyclohexylamino)-2,3-dihydro-1*H*-inden-1-one (BCI) induced both elimination of CML stem and progenitor cells assayed in LTC-IC while sparing normal CD34^+^. A total of 90% of mice that belong to CML mouse models were recovered without identifying any residual TKI-resistant LSC [133]. Another important result, achieved by Kesarwani et al. emphasized the important role of the *DUSP* family. By analyzing different interactions between MAPKs and DUSPs, they show that in addition to the TKI resistance induced by the overexpression of DUSP1, *DUSP6*^−/−^-deficient cells are also TKI-insensitive, demonstrating the critical role played by these different PPs in TKI responsiveness [133]. Another PP, *PTPRG*, has been validated as a regulator of the response to second-generation TKIs in a newly diagnosed cohort of 66 CML patients, with the detection of higher mRNA level after 9 months of nilotinib treatment. Very recent data have confirmed the restorations of PTPRG after TKIs treatment in CML patients, suggesting PTPRG as a possible tool to monitor response to TKIs [134]. Overall, PTPRG expression positively correlated with the response to second-generation TKIs. This occurrence was independently confirmed in K562 cells, where, overexpression of PTPRG significantly improved the response to nilotinib and decreased its IC_50_ [135]. Other studies have focused on the opposite dual role assigned to the SH2-containing protein tyrosine phosphatases SHP1 (*PTPN6*) and SHP2 (*PTPN11*) in promoting IM resistance. Epigenetic processes were shown to regulate SHP1 mRNA expression in CML and, when these data were compared to healthy individuals, SHP1 expression levels were found to be downregulated at diagnosis and further reduced during disease progression phases (AP and BP) [136,137]. Moreover, SHP1 phosphatase appears to be downregulated in non-responding CML patients and TKIs resistant cell lines, strongly limiting its interaction and regulation of SHP2 phosphatase activity, which has already been implicated in TKI resistance when its interaction domain on GAB2 is mutated [138]. Conceivably, as has been further demonstrated for another TKI-resistant CML cell line [139], SHP1 was downregulated in a cohort of CML patients that failed IM treatment. Its role as SHP2 phosphatase was to act on the tyrosine residue Y542, defining its capacity to control activators signals SHP2 receives from BCR-ABL1 and membrane receptor kinases. This establishes BCR-ABL1-independent regulation of IM resistance [44]. PP2A-mediated regulation of SHP1 phosphatase signaling constitutes an additional molecular pathway, which is deeply involved in the lack of effectiveness of TKI treatments [96]. PP2A-activating drugs (PADs)-mediated SET disruption and reactivation of PP2A (e.g., forskolin or FTY-720) have been shown to cause caspase-dependent apoptosis in primary Ph^+^ cells [98,140]. In addition, TGI1002 containing 2-phenoxypyrimidine or OP449 peptide directly interacts with SET, with a dose-dependent efficacy in reducing Y307 phosphorylation on the catalytic sub-unit of PP2A [141]. Phosphatase restoration by these compounds contributes to enhanced TKI response (as in the case of OP449), as well as allowing the overpass of resistance even in those cells carrying a drug-resistance mutation in the BCR-ABL1 kinase domain, as has been shown in primary CD34^+^ CML cells or in a mouse models implanted with drug-resistant Ba/F3-p210^T315I^ Bcr-Abl1 [142]. In support of these findings, the involvement PP2A in drug-tolerance is governed by further signaling pathways. S1P2-mediated sphingosine kinase-1 (SK-1)/sphingosine 1-phosphate (S1P) activation constitutes a regulatory pathway for this phosphatase. Early data have indicated strong C18-ceramide production in response to IM-treatment, thought an increase in expression levels of sphingosine kinase-1 (SK-1) and the generation of sphingosine 1-phosphate (S1P) in the K562 IM-sensible cell line [143]. This impaired cascade prevents BCR–ABL1 dephosphorylation and degradation by improving its stability through inhibition of the PP2A/SHP1 axis [144]. Pharmacological or genetic hindrance of the SK-1/S1P2 pathway can increase IM- or Nilotinib-response in primary CD34^+^ CML progenitors. On the other side, NADPH oxidase-guided ROS production induce PP2A abrogation. Imipramine blue (IB), a specific NADPH oxidase inhibitor, lessened the survival of CML LSC, reinforcing nilotinib action through ROS reduction, a mechanism well-known to inactivate protein phosphatases through sulfhydryl group oxidation [145]. Hence, IB seems to positively modulate PP2A activity, allowing it to exert its anti-leukemic activity. Lai et al., reporting data on PIDs (PP2A-inhibiting drugs) such as LB100 (or the lipid-soluble LB102), have recently reconsidered this view on PP2A role in CML. These drugs seem able to prevent PP2A-dependent dephosphorylation on β-catenin, specifically on their S/T residues in N-terminal domain. Indeed the synergic effect induced with a TKI-treatment avoided the stabilizing action of BCR-ABL1 on Y-residues of β-catenin, inducing proteasome degradation and decreasing the expression of several downstream genes (e.g., TCF1/LEF1, CCND1, cMYC). Furthermore, NSG mice transplanted with aggressive BCR-ABL^+^ blast cell (BV173) were treated with vehicle, DAS, LB100 and their combination for two weeks. Three weeks after treatment discontinuation spleen and liver enlargement was prevented and leukemia cells infiltration was undetectable only in the case of combination of DAS plus LB100 [146]. However issues regarding specificity of the drugs employed, in vivo off-target mediated toxicity of LB100 and lack of validation in human primary cell models were raised [147,148]. Targeting LSCs has been the focus of other studies considering the tyrosine phosphatase family, where a fundamental role is played by ICSBP-mediated PTPN13 (FAP-1) enhancement. In human CP-CML CD34^+^ cells, FAP-1 and its target β-catenin proteins have been shown to be upregulated, compared to that in control subjects. SLV peptide (FAP-1 inhibitor) treatment abrogated β-catenin accumulation, an observation corroborated also by the reduction in its target genes (e.g., cMYC and SURVIVIN) [109,110]. Furthermore, Fap-1 phosphatase activity induced leukemia recurrence at 18 weeks after treatment with IM in Bcr-Abl1-transduced mice. Conversely, treatment with Fap-1 inhibitors, such as SLV peptide or a Quinobene inhibitor, in the presence of IM has been associated with 100% survival after 32 weeks without relapse in secondary recipient Bcr-Abl1-transduced mice. Of relevance, tertiary recipient mice, inoculated with BM cells from previous IM + Fap1 inhibitor-treated secondary recipients mice, preserved normal hematological parameters after 24 weeks without any additional treatments, contrary to tertiary recipients receiving BM cells from other treatment and control vehicle mice, which demonstrated CP relapse and progression to BP [149]. Hence, FAP-1 might represent a leading factor driving TKI resistance.

## 3. Pathogenetic Role of Protein Phosphatases Requiring Validation in Primary CML Cells or Mouse Leukemia Models

An analysis of the literature concerning PPs revealed that some have shown great significance in CML; however, others lack validation in primary patient-derived samples or CML mouse-models (as summarized in Table 1). Among these is PTP1B, a non-receptor phosphatase encoded by the *PTPN1* gene. PTP1B is a key regulatory molecule, which has been shown to antagonize several tyrosine kinases and appears to be implicated in different signaling pathways, such as those related to insulin resistance, immune response (regulatory role in dendritic cells) [150], and on the development of many cancers. Indeed, *PTP1B^−/−^* mice suffered a shorter lifespan, through the development of acute myeloid leukemia [151]. PTP1B phosphatase is able to bind BCR-ABL1 and seems to play a tumor-suppressing role in CML cell lines; especially by reducing their ability to form colonies and viability [152,153]. Reduced activity or expression of PTP1B was associated with ubiquitination-mediated BCR-ABL1 protein degradation, implying a protein-stabilizing role of this phosphatase [152]. Equally, the inhibition of PTP1B activity causes IM resistance, through an increase in kinase activity of BCR-ABL1 and blockage of the IM-induced effects in K562 and in SupB-15 (Ph^+^ ALL cell line). Nevertheless, conflicting data are available on the positive and negative roles of PTP1B in the IM-mediated response [152,154,155]. Another cytoplasmic PTP is LYP (*PTPN22*). This phosphatase has been detected specifically in lymphocytes, but also described in numerous myeloid cell lines, including normal circulating myeloid cells. Over-expression of LYP in a CML cell line, KCL22, demonstrated its interaction with BCR-ABL1 and its downstream target proteins, such as GRB2 protein, thus mitigating BCR-ABL1 leukemogenesis. Furthermore, LYP may dephosphorylate other BCR-ABL1 substrates such as CBL, CRKL, and ERK1/2, suggesting that it may play an important role in CML. However, mRNA expression data from primary samples derived in different stages of CML did not agree with the results obtained in KCL22 cell line, demonstrating a need for further investigation [156]. Remaining in the PTP family the receptor type, f polypeptide, leukocyte common antigen (LAR) interacting protein (liprin), alpha 1 (*PPFIA1*) has been shown to be downregulated by direct 3′ UTR binding of miR-181a in CML. Using an miR-181a mimic and *PPFIA1* siRNA transfection, an increased IM sensitivity appeared with a clear K562 growth reduction that was promptly reversed in the case of PPFIA1 overexpression. Imatinib-treated PPFIA1-overexpressing CML cells considerably inhibited the cell death effect of the TKI [157]. As previously mentioned for PP2A phosphatase, the increased ROS concentration produced by BCR-ABL1 in CML LSC causes its inactivation. This appears to be the case for PP1α (*PPP1CA*), another Ser/Thr-specific phosphatase whose alpha sub-unit of the PP1α complex associates with over 200 regulatory proteins to form holoenzymes which dephosphorylate their biological targets with high specificity, thus playing crucial roles in the regulation of numerous cellular processes [21,158]. Particularly, Naughton et al. identified a class of NADPH oxidases (Nox-4) whose activity is induced by BCR-ABL1 expression. Enhanced ROS production observed in TonB.210—a BCR-ABL1-expressing cell line—determined the redox inhibition of PP1α. Inhibition of this phosphatase, in turn, induces the PI3K/AKT pathway, leading to improved cell survival and resistance to apoptosis, thus emphasizing the role played by this phosphatase in CML [159]. Low molecular weight protein tyrosine phosphatase LMW-PTP (*ACP1*) is an 18-kDa enzyme, whose upregulation is often associated to increased cellular proliferation in several malignancies [160]; for instance, in colorectal cancer, it plays an oncogenic function and has been proposed as a predictive biomarker. In CML, this phosphatase appears to be involved in the regulation of autophagy processes [161]. Furthermore, using Lucena-1 (K562 multi-drug resistant cell line over-expressing LMWPTP), a relationship to IM-resistance was shown, the mechanism of which appears to be related to the maintenance of an activated state of several Src-kinases, including BCR-ABL1 [162]. Another PP, TC-PTP (*PTPN2*), is expressed in two differentially spliced isoforms (TC45 and TC48) and has been shown to have a role in the hematopoietic system and in inflammatory responses [163]. TC45 phosphatase is typically expressed in the nucleus, while TC48 only in the cytoplasm. TC45 has been shown to be involved in the response to INF-α and resistance to IM in a KT-1 cell line [164]. In particular, downregulation of TC45 is related to a marked expression of STAT1 and STAT5 phosphorylated proteins, as identified in TKI-resistant KT-1 sub-clones. Forced re-expression of TC45 in these sub-clones did not produce a direct effect on the phosphorylation status of the JAK family kinase, BCR-ABL1, or other downstream effectors; instead, this phosphatase induced: (1) A recovery of IM sensitivity by reducing pSTAT5 stimulating apoptosis, and blocking proliferation [165]; and (2) a specific resistance to INFα, with a mechanism independent of TKI-resistance [164]. Both results suggest the role of TC45 in the regulation of p-STAT proteins within the nucleus. Contrarily, the TC48 cytoplasmic form acts directly on the BCR-ABL1 protein by binding with its BCR domain, thus producing dephosphorylation and reducing its activity. As in the nuclear context, expression of TC48 variant may reduce cell proliferation and increase the effects of IM [166].

## 4. Conclusions

Despite a dramatic improvement of patient survival, a final cure achievable through the complete eradication of CML cells by targeted therapy remains out of sight for a majority of patients. The additional pathways which contribute to controlling the pathogenetic process active in CML need to be uncovered in details, in order to identify the relevant pathways that must be targeted in order to obtain this goal, which seem closer for CML than for many other types of leukemia. Recent work has delineated differential pathways governed by p210 and p190 BCR-ABL1, suggesting an unexpected and complex variety of signaling events governed by this oncogene [28,167]. Given these premises, not only kinase inactivation but also its degradation might represent another way to tackle the ineffectiveness of and resistance to TKIs. An example of this approach might be an SPOA engineered system created by fusing GRB2 SH2-domain, PTP1B/C domain and ornithine decarboxylase (ODC, a specific label which has been shown to be effective), together with antizyme AZ1, to specifically dephosphorylate and degrade the BCR-ABL1 protein [168]. SPOA strongly deactivates BCR-ABL1 Y177, preventing GRB2 binding and blocking proliferation, as well as inducing apoptosis in even IM-resistant CML cells [169]. Moreover, emerging new disease models such as two recently developed humanized zebrafish expressing BCR-ABL1, might provide future instruments to refine our understanding of the disease and the roles of phosphatases in CML [170,171]. In this review, we attempted to summarize the current findings defining the molecular events governing CML, through the relationship between kinases and phosphatases. Most of these pathways inevitably involve the most relevant oncogenic driver in CML, BCR-ABL1, whose regulatory mechanisms—despite the wealth of data produced over the last decades—are still worth investigation. As an example, no literature data are available on the direct measurement of BCR-ABL1 kinase activity in primary Ph^+^ leukocytes, as only indirect assay have been described [172,173,174,175,176]. The data here discussed indicate how some of these mechanisms are independent from the BCR-ABL1 oncogene, thus representing interesting targets for the development of specific agonists or antagonists. Phosphatases together with the class of Non-coding RNA, such as MicroRNAs are very significant players in CML. Recently, miR-15/16 loci, known to be downregulated in CML patients and in other leukemia types [177,178], were found to be correlated with the progression to the BP of CML [179]. Phosphatases, miRNAs, and related nucleic acid-based regulators are emerging new players, which we expect to become targetable in the near future and, thus, may be tested together with TKIs, in order to target the diverse and complementary pathways that support the viability of CML cells (and stem cells).

## Figures and Tables

**Figure 2 cancers-13-02311-f002:**
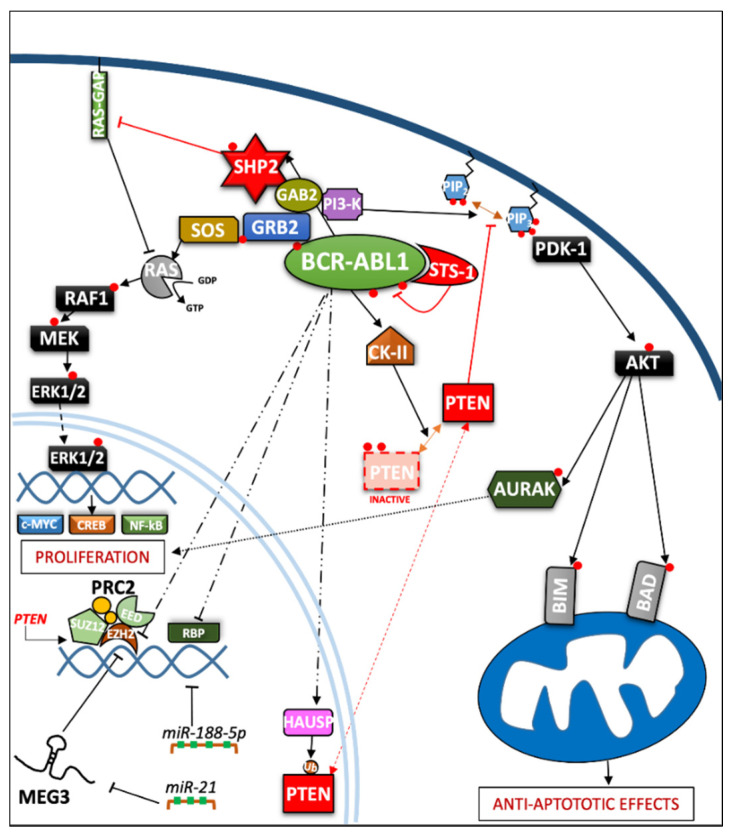
Depiction of the BCR-ABL1 kinase-driven proliferative processes in the context of CML. ERK = extracellular signal-regulated kinases; MEG3 = maternally expressed gene 3; PRC2 = polycomb repressive complex 2; EZH = enhancer of zeste polycomb repressive complex sub-unit.

**Figure 3 cancers-13-02311-f003:**
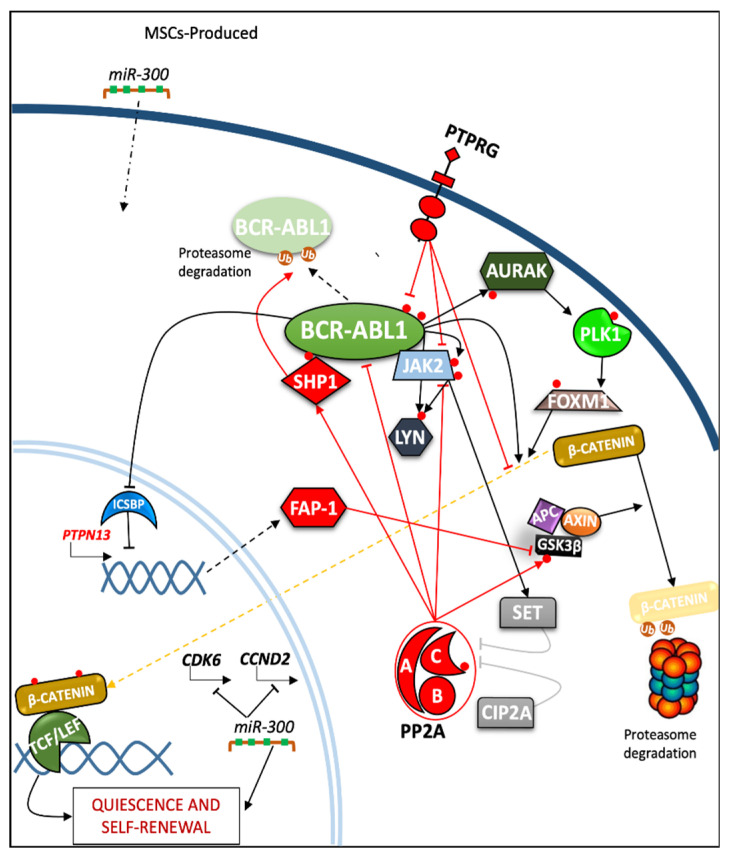
The major mechanisms involving phosphatases in self-renewal and quiescence processes in CML cells. FOXM1 = forkhead box protein M1; TCF = transcription Factors; LEF = lymphoid enhancer-binding factor 1.

**Figure 4 cancers-13-02311-f004:**
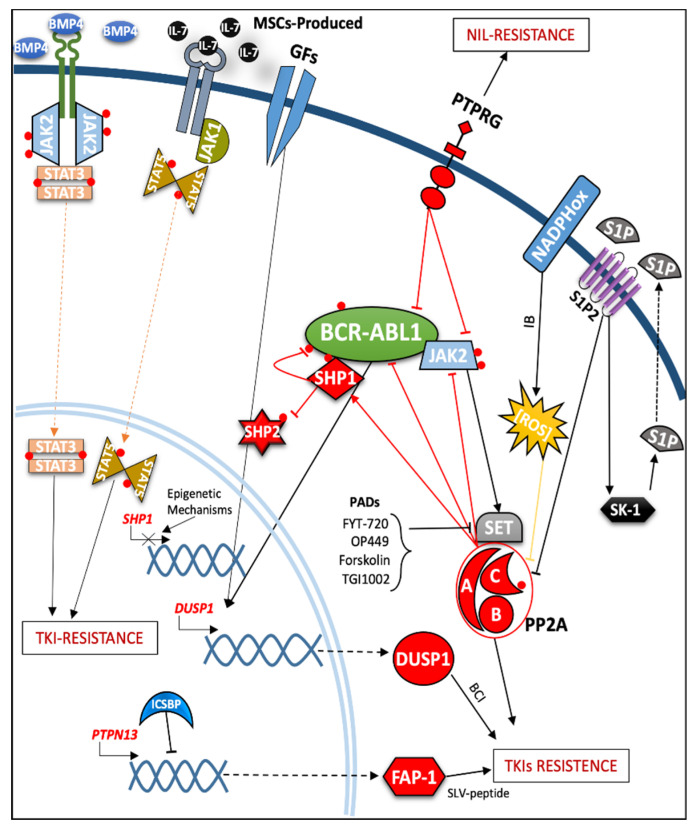
Schematic representation of the role played by kinase and phosphatases in the resistance or sensitization mechanisms to TKIs in CML. GFs = growth factors; ROS = reactive oxygen species; PADs = PP2A activating drugs; NADPHox = nicotinamide adenine dinucleotide phosphate oxidase; MSCs = mesenchymal stem cells; BMP4 = bone morphogenetic proteins; STAT = signal transducer and activator of transcription.

**Table 1 cancers-13-02311-t001:** PPs in CML.

Protein	Coding Gene	Role in CML	References
Data verified in primary CML cells or in leukemia mouse models			
DUSP1	*DUSP1*	Implicated in TKI-response	[133]
STS-1	*UBASH3B*	Decreases cell proliferation Direct BCR-ABL1 regulation	[27,30]
FAP-1	*PTPN13*	Regulation of β-catenin functions Decreases TKI sensitivity	[105,106,107,109,110,149]
PTPRG	*PTPRG*	Regulation of β-catenin functions Implicated in TKI response	[112,113,114,116,117,118,120,134,135]
SHP1	*PTPN6*	Acts through the PP2A on BCR-ABL1 Regulates BCR-ABL1-independent IM resistance	[44,96,97,136,137,144]
SHP2	*PTPN11*	Increases cell proliferation Implicated in TKI resistance	[35,37,38,39,40,41,42,43,44]
PP2A	*PPP2CA*	Quiescence and Self-renewal regulation Governs TKI-response	[96,97,98,99,100,103,140,141,142,144,145,146,147,148]
PTEN	*PTEN*	Control of cell proliferation	[49,59,60,61,63,64,66,67,68,70,71]
Data obtained only in CML cell lines			
PTP1B	*PTPN1*	Reduces cell viability Correlated with IM response	[150,151,152,153,154,155]
LAR –LIPRIN Α1	*PPFIA1*	Mitigates BCR-ABL1 leukemogenesis	[157]
LYP	*PTPN22*	Decreases IM sensitivity	[156]
LMW-PTP	*ACP1*	Regulates autophagy process Correlated with IM resistance	[161,162]
PP1Α	*PPP1CA*	Improves cell survival and apoptosis resistance	[159]
TC45/TC48	*TC-PTP*	Implicated in IM- and INFα-resistance Regulation of proliferation and apoptosis	[164,165,166]

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
