# Peer review of "Current Views on the Interplay between Tyrosine Kinases and Phosphatases in Chronic Myeloid Leukemia"

_cancers, 2021, doi:10.3390/cancers13102311_

Round 1

Reviewer 1 Report

Comments and Suggestions for Authors

The story of Boni et al. is a comprehensive review on the interplay between tyrosine kinases and phosphatases in Chronic Myeloid Leukemia. “Unfortunately, we are still far away from the eradication of LSCs, despite the long survival with a good quality of life in more than 90% of patients affected by CML”.

The paper is well written (some grammatical checks are required) and informative; as such it deserves the publication in Cancers. Below, I enclose a few comments that can potentially help to further improve/strengthen its message: 

The distinction between leukemia-initiating cells and leukemia stem cells is a matter of many misunderstandings. As the Authors comprehensibly cover this point, I would suggest the introduction of the diagram that would depict the differences between LICs and LSCs;

The authors should include the concept of "clonal evolution of LSCs" and it should be emphasized as a consequence of LSC drug-resistance/heterogeneity;

The use of many abbreviations in some points make the manuscript difficult to read. I recommend including a list of all abbreviations used in the text and paying attention to write the full names of the acronyms reported in the text when spelled for the first time;

LSCs in CML have been universally identified as CD34+CD38-CD90+CD26+. The authors should include the LSCs markers definition in the section 2.2;

The authors should also define better what the DLK1-DIO3 imprinted locus is.

Author Response

Thanks to all the reviewer for the helpful suggestions. English was professionally edited by a dedicated service provided by MDPI, beside we have specifically addressed the points raised by the reviewers. In yellow are the parts added.

Suggestion: 1: The distinction between leukemia-initiating cells and leukemia stem cells is a matter of many misunderstandings. As the Authors comprehensibly cover this point, I would suggest the introduction of the diagram that would depict the differences between LICs and LSCs;

Response:  In paragraph 2.2 lines 283-305 we have expanded and better defined the features of these two important cell populations

Suggestion 3: The use of many abbreviations in some points make the manuscript difficult to read. I recommend including a list of all abbreviations used in the text and paying attention to write the full names of the acronyms reported in the text when spelled for the first time;

Response:  To better clarify all the acronyms used in the paper, we have added a list of acronyms after the conclusion paragraph and under each figures.  Each mentioned acronym was described in its extended form the first time it appeared in the text.

Suggestion 4: LSCs in CML have been universally identified as CD34+CD38-CD90+CD26+. The authors should include the LSCs markers definition in the section 2.2;

Response:  Thanks to the reviewer for the suggestion. We have added the profile of the surface markers indicated for LSC cells (Paragraph 2.2 line 299-302).

Suggestion 5: The authors should also define better what the DLK1-DIO3 imprinted locus is.

Response:  Thanks to the reviewer for the suggestion. In lines 351-355 we briefly described the imprinting that can regulate the genes present on the genomic region DLK1-DIO3

Reviewer 2 Report

Boni and Sorio have reviewed in this manuscript the interplay between the fusion tyrosine kinase, BCR-ABL1, and different phosphatases in Chronic Myeloid Leukemia (CML). The authors made an exhaustive description of what is currently known on phosphatases activities in CML. I have some minor comments to make the review easier to digest as it can be quite dense.

  1. The authors should add a figure showing a scheme of BCR, ABL, and BCR-ABL with the different domains, post-translational modifications cited in the manuscript, and possibly the direct interacting partners.
  2. I would recommend to divide the current figure 1 in three figures, one for each part. This will also allow to make a few panels for each new figure where standalone regulatory pathways could be put to make the figures easier to follow.
  3. Some of the acronyms need to be defined or made consistent: PTKRs, PTKNRs, RPTKs, NRTKs (lines 49-53); HSCs (line 75), ALL (line 136), LSCs (line 230).
  4. The English will need to be improved.

Author Response

Thanks to all the reviewer for the helpful suggestions. English was professionally edited by a dedicated service provided by MDPI, beside we have specifically addressed the points raised by the reviewers. In yellow are the parts added.

Suggestion 1: The authors should add a figure showing a scheme of BCR, ABL, and BCR-ABL with the different domains, post-translational modifications cited in the manuscript, and possibly the direct interacting partners.

Response:  We added Figure 1 to describe the individual products of the BCR and ABL1b genes and the altered version of the BCR-ABL1 tyrosine kinase. The main interacting proteins of BCR-ABL implicated in leukemogenesis are represented.

Suggestion 2: I would recommend to divide the current figure 1 in three figures, one for each part. This will also allow to make a few panels for each new figure where standalone regulatory pathways could be put to make the figures easier to follow.

Response:  We followed the advice and the previous image was separated into 3 different images by dividing them into each specific paragraph.

Suggestion 3: Some of the acronyms need to be defined or made consistent: PTKRs, PTKNRs, RPTKs, NRTKs (lines 49-53); HSCs (line 75), ALL (line 136), LSCs (line 230).

Response:  To better clarify all the acronyms used in the paper, we have added a list of acronyms after the conclusion paragraph and under each figures.  Each mentioned acronym was described in its extended form the first time it appeared in the text.

Suggestion 4: The English will need to be improved.

Response: We used the English review service provided by the journal to improve the level of English.

Reviewer 3 Report

Dr. Boni and Sorio presented an interesting and completed review of the most important evidences concerning the interplay between tyrosine kynases and phospatases in human CML.

The manuscript is well organized, even if I have some concerns/comments.

  • In the Introduction section, human CML and the leukemogensis sustained by BCR-ABL1 is presented twice (Lines 61 and 76). It is quite confusing. I strongly suggest the Authors to reorganize the Introduction section avoiding anticipations or repetitions.
  • The sentence from line 454 to line 458 is quite long and complicated. Please, rephrase it.
  • Line 465 "NGS mice injected with BCR-ABL+ blast cell (BV173) the treatment with combination of DAS and LB100...". I think that the sentence lacks of some words or verbs. Please, rephrase the sentence or reorganize it.
  • Section number 3. The Authors have presented the most important studies/results about the investigation of the role of phosphatases in CML requiring validation in primary CML cells or in mouse models. Recently, two different human CML zebrafish models have been presented (PMID 31289206 and PMID 33669758). Despite no results have been yet published about phosphatases in CML zebrafish models, it is undoubtedly that zebrafish presents many advantages than mouse from several point of view. I suggest the Authors to introduce these two zebrafish models and to comment their hypothetical involvement in further studies at the end of section 3 or in the discussion as future perspectives.
  • There are some acronyms not defined: Abstract, line 25 "PPs", Introduction, lines 49 and 53 "RPTKs" and "NRTKs", respectively.
  • There are several typo errors and I also suggest to uniform the verbal tenses. Please, see as examples line 325 "act" instead of "acts" (the same at line 571), line 327 "downregulate" instead of "downregulates", line 355 "occupy" instead of "occupies", line 387 "associate" instead of "associated", line 391 "affect" instead of "affects", line 399 "induce" instead of "induces or induced", line 415 "cell" instead of "cells" and a comma after "where", line427 "define" instead of "defines", line429 "establish" instead of "establishes", line 430 "constitute" instead of "constitutes" (the same at line 442), line 437 is quite confusing. Maybe "allowing the overpass of" sounds better?

I hope my suggestions and comments will help the Authors in improving the quality of this very interesting Review.

Author Response

Thanks to  the reviewer for the helpful suggestions. English was professionally edited by a dedicated service provided by MDPI, beside we have specifically addressed the points raised by the reviewers. In yellow are the parts added.

Suggestion 1: In the Introduction section, human CML and the leukemogensis sustained by BCR-ABL1 is presented twice (Lines 61 and 76). It is quite confusing. I strongly suggest the Authors to reorganize the Introduction section avoiding anticipations or repetitions.

Response: We rephrased the sentence that was, indeed,  quite confusing by removing the repetition.

Suggestion 2: The sentence from line 454 to line 458 is quite long and complicated. Please, rephrase it.

Response: We totally agree and have rewritten the whole chapter focusing on the discussion related to the opposite role of PP2A proposed and subject of debate in the literature (now 501-513), please advise.

Suggestion 3: Line 465 "NGS mice injected with BCR-ABL+ blast cell (BV173) the treatment with combination of DAS and LB100...". I think that the sentence lacks of some words or verbs. Please, rephrase the sentence or reorganize it.

Response: We rephrased the sentence to make it more understandable, please advise

Suggestion 4: Section number 3. The Authors have presented the most important studies/results about the investigation of the role of phosphatases in CML requiring validation in primary CML cells or in mouse models. Recently, two different human CML zebrafish models have been presented (PMID 31289206 and PMID 33669758). Despite no results have been yet published about phosphatases in CML zebrafish models, it is undoubtedly that zebrafish presents many advantages than mouse from several point of view. I suggest the Authors to introduce these two zebrafish models and to comment their hypothetical involvement in further studies at the end of section 3 or in the discussion as future perspectives.

Response: We were aware of these two new Zebrafish animal models we originally did not include as they are still in the modelling phase but we agree it is worth mentioning. In the conclusions we included and commented on the use of this animal model as a promising for future studies on CML pathways. (lines 618-621)

Suggestion 5: There are some acronyms not defined: Abstract, line 25 "PPs", Introduction, lines 49 and 53 "RPTKs" and "NRTKs", respectively.

Response:  To better clarify all the acronyms used in the paper, we have added a list of acronyms after the conclusion paragraph and under each figures.  Each mentioned acronym was described in its extended form the first time it appeared in the text.

Suggestion 5: There are several typo errors and I also suggest to uniform the verbal tenses. Please, see as examples

  • line 325 "act" instead of "acts" (the same at line 571),
  • line 327 "downregulate" instead of "downregulates",
  • line 355 "occupy" instead of "occupies",
  • line 387 "associate" instead of "associated",
  • line 391 "affect" instead of "affects",
  • line 399 "induce" instead of "induces or induced",
  • line 415 "cell" instead of "cells" and a comma after "where",
  • line427 "define" instead of "defines",
  • line429 "establish" instead of "establishes",
  • line 430 "constitute" instead of "constitutes" (the same at line 442),
  • line 437 is quite confusing. Maybe "allowing the overpass of" sounds better?

Response:  We corrected the typos and forwarded the ms to the English review service provided by the journal to improve the level of English.

Round 2

Reviewer 3 Report

The Authors really improved the readability and the quality of the Manuscript and replayed to all my comments/suggestions.